# Forensic DNA Analysis of Mixed Mosquito Blood Meals: STR Profiling for Human Identification

**DOI:** 10.3390/insects14050467

**Published:** 2023-05-16

**Authors:** Ashraf Mohamed Ahmed, Amani Mohammed Alotaibi, Wedad Saeed Al-Qahtani, Frederic Tripet, Sayed Amin Amer

**Affiliations:** 1Zoology Department, College of Science, King Saud University, Riyadh 11451, Saudi Arabia; 2Department of Forensic Sciences, College of Criminal Justice, Naif Arab University for Security Sciences, Riyadh 11587, Saudi Arabia; 4380665@nauss.edu.sa (A.M.A.); wqahtani@saip.gov.sa (W.S.A.-Q.); 3Swiss Tropical and Public Health Institute, Kreuzstrasse 2, 4123 Allschwil, Switzerland

**Keywords:** mosquitoes, *Culex pipiens*, mixed blood meals, post-feeding, forensic, human identification, STR profiling

## Abstract

**Simple Summary:**

Blood-fed mosquitoes found at crime scenes can provide human blood traces for genetic analysis that aid in forensic investigations. Because a mosquito can obtain its blood meal from two or more sources, this study aims to determine whether it is possible to obtain a human DNA profile from the blood meal of a single *Culex pipiens* mosquito experimentally fed on mixed blood of human male-female or human-mice. Twenty-four human STR loci were amplified over a complete digestion profile post-feeding. Data revealed that full, complete, and partial STR profiles could be obtained from all blood meal types up to 12, 24, and 36 h post-feeding, respectively. The mixed human–animal blood maximized the DNA degradation and, thus, impaired STR identification beyond 36 h until they became poorly detectable at 48 h post-feeding. These results indicate that human DNA can be obtained from mosquito blood meals, even if it is mixed with non-human blood, for up to 36 h post-feeding.

**Abstract:**

Mosquito vectors captured at a crime scene are forensically valuable since they feed on human blood, and hence, human DNA can be recovered to help identify the victim and/or the suspect. This study investigated the validity of obtaining the human short tandem repeats (STRs) profile from mixed blood meals of the mosquito, *Culex pipiens* L. (Diptera, Culicidae). Thus, mosquitoes were membrane-feed on blood from six different sources: a human male, a human female, mixed human male-female blood, mixed human male-mouse blood, mixed human female-mouse blood, and mixed human male-female-mouse blood. DNA was extracted from mosquito blood meals at 2 h intervals up to 72 h post-feeding to amplify 24 human STRs. Data showed that full DNA profiles could be obtained for up to 12 h post-feeding, regardless of the type of blood meal. Complete and partial DNA profiles were obtained up to 24 h and 36 h post-feeding, respectively. The frequencies of STR loci decreased over time after feeding on mixed blood until they became weakly detectable at 48 h post-feeding. This may indicate that a blood meal of human blood mixed with animal blood would contribute to maximizing DNA degradation and thus affects STR identification beyond 36 h post-feeding. These results confirm the feasibility of human DNA identification from mosquito blood meals, even if it is mixed with other types of non-human blood, for up to 36 h post-feeding. Therefore, blood-fed mosquitoes found at the crime scene are forensically valuable, as it is possible to obtain intact genetic profiles from their blood meals to identify a victim, a potential offender, and/or exclude a suspect.

## 1. Introduction

Violent crimes are committed all over the world and usually result in bloodshed. The accurate detection and confirmation of human blood traces and bloodstains obtained from crime scenes are crucial in forensic analysis [1]. The information gained from the correct analysis of bloodstains can include what did (or did not) take place, answer the question of who may have been involved in these actions, and could be the determining factor between guilt and innocence. Blood is usually found at the crime scene as dried stains on clothes or solid surfaces, such as wood, knives, brick, etc., which need specific materials and methods for proper collection and identification [2]. In comparison, a blood-fed mosquito (if found at the crime scene) can provide a fresh human blood sample (its blood meal) that is suitable for forensic analysis [3]. Adult female mosquitoes of the genera *Aedes*, *Culex*, and *Anopheles* (Family Culicidae) feed on human and animal blood to obtain essential nutrients needed for vitellogenesis (egg production) [4,5,6]. After blood feeding, mosquitoes tend to stay for a few hours near the site where they took their blood meals [6,7,8] and prefer to take another blood meal a couple of hours later [5]. This particular feeding behavior makes mosquitoes potentially useful evidence if found at the crime scene, as their blood meal can be considered a human trace sample; hence, it is a real forensic investigation tool [9,10].

Several previous studies have observed and documented mosquitoes at crime scenes in different countries [11,12,13]. These studies provided evidence that blood-fed mosquitoes found at crime scenes can be used as a valuable source of forensic information that could solve a case, as they could have fed on the blood of a suspect or victim [11]. Thus, human DNA profiles can be determined from the mosquito blood meal, which facilitates the identification of a victim at the crime scene or determines whether a potential suspect was present nearby. 

The analysis of short tandem repeats (STRs) found on human DNA obtained from blood-fed mosquitoes is currently a new approach used in forensic investigations in order to determine the identity of individuals [14]. These STRs are located in non-coding regions of the DNA, and each of them is composed of repeated units seen in tandem at a particular location on a chromosome [15], and consequently, they can be used for generating a characteristic DNA profile for a person [14,15]. Several previous studies have shown the possibility of obtaining forensically valuable human DNA profiles from blood meals of various mosquito species up to 48 h after the blood meal. These DNA profiles were successfully used to determine the identity of the persons from whom the blood meals were taken [3,9,12,13,16,17].

However, there are two major limitations to using mosquitoes as a source of human DNA at the crime scene that should be taken into account: (a) the DNA degradation that occurs after a blood meal by the digestion process that can last 60–70 h [18]. In this regard, several studies have investigated the effect of post-feeding time on generating full STR profiles from DNA extracted from mosquito blood meals, ranging from 24 to 48 h, with the possibility of performing successful human identification up to 72 h post-feeding [3,12,19]; (b) DNA extraction and quantification from mosquito blood meals is time-consuming, labor-intensive, expensive, and consumes all of the mosquito samples. However, this issue has been mitigated by using direct amplification methods to produce DNA profiles within a short period of time [20,21].

Biological traces detected at a crime scene may contain mixtures of two or more sources and therefore have forensic value [22]. The objective of this study was to explore the capability of obtaining STR profiles on human DNA recovered from blood meals of the mosquito, *Cx. pipiens* L., upon feeding on mixed human and animal blood. The Swiss albino mouse, *Musculus domesticus*, was used in this study as a source of animal blood. Non-mixed blood meals taken from humans, male or female, or mixed with mice blood were used for DNA profiling over the digestion period post-feeding.

## 2. Materials and Methods

### 2.1. Experimental Mosquitoes

A stock colony of *Cx. pipiens* L. mosquitoes were reared in the insectary of the Zoology Department, College of Science, King Saud University, under standardized conditions according to [23]. Briefly, 6-day-old adult female mosquitoes were starved for 12 h prior to blood feeding on a healthy male Swiss albino mouse obtained from the Animal House, Zoology Department, College of Science, King Saud University. The mouse was anesthetized by intraperitoneal injection with TEKAM 50 (Pharmaceuticals, Amman, Jorden) prior to the mosquito blood feeding, which lasted for 20–30 min. Three days later, plastic pots filled with distilled water were provided for mosquito egg-laying. Each of the 200 newly hatched larvae was distributed into 1 L of distilled water in plastic trays (34 × 24 × 10 cm^3^ each). One drop of Liquifry (Interpet Ltd., Dorking, UK) was added to each dish for the first two days, and then ground Goldfish Flake Food (Warldley^®^, The Hartz Mountain Corporation, Secaucus, NJ, USA) was provided until pupation. The resulting pupae were separated daily, placed in distilled water-filled polystyrene pots, and kept in their rearing wooden cages (30 × 30 × 30 cm^3^) until the adult emergence of the next generation. The resulting adult mosquitoes were allowed permanent access to a 10% glucose solution (*w*/*v*) ad libitum and were continuously reared in adequate numbers for the experimental purposes of this study.

### 2.2. Experimental Mice

The Swiss albino male mouse, *Musculus domesticus*, was tested, housed, and handled according to the Animal Research: Reporting of In Vivo Experiments (ARRIVE) and Federal Animal Welfare guidelines. All animals were housed in the breeding rooms of experimental animals under constant temperature (20–22 °C) in the Animal House of Zoology Dept., College of Science, King Saud University. Briefly, a good-sized plastic cage (25 × 22 × 30 cm^3^) with wires coated with a metal mesh with holes for water and food supply were cleaned twice a week. The cages had translucent walls, which enable experimenters to observe them. Animals were given ad libitum access to food from the General Company of Oils and Grain in Riyadh. All mice were maintained under a 12 h light: 12 h dark photocycle. 

### 2.3. Blood Samples

#### 2.3.1. Human Blood

All human donors voluntarily provided written permission to participate in the study and to obtain their blood samples for research purposes. Healthy adult human donors included a 21-year-old male and a 40-year-old female of similar blood groups (O^+^) and participated in this study after providing informed consent. Blood samples (4 mL each) were collected from arm veins into a 4.0 mL Lithium Heparin tube (Sterilite^®^, Townsend, MA, USA) (www.sterilite.com, accessed on 12 May 2023) and immediately used for membrane feeding of the experimental mosquitoes.

#### 2.3.2. Mice Blood

The experimental Swiss albino mouse (≈25 g, 12–14 weeks old) was used in this study as a source of animal blood. Blood was freshly collected by cardiac puncture, as detailed in List [24], prior to experimental use. Briefly, the mouse was anesthetized by intraperitoneal injection with TEKAM 50 (Pharmaceuticals, Amman, Jorden). Under anesthesia, the mouse was placed in dorsal recumbency, and an appropriately sized needle of a heparinized syringe was directed and inserted toward the animal’s heart (held 20–30 degrees off horizontal). Up to 0.75 mL of blood can be collected from one mouse of 25 g weight [24]. Collected blood samples from five mice were immediately pooled into a 4.0 mL lithium heparin tube. Prior to experimental use, blood-tubes were kept in a water bath at 37 °C for 5 min as recommended by [25].

### 2.4. Mosquito Membrane Feeding

Mosquitoes were allowed to feed on the experimental blood using the Hemotek 5W1 Membrane Feeding System (Hemotek, Blackburn, UK) (http://hemotek.co.uk/products/, accessed on 12 May 2023) according to the manufacturer’s instruction manual as detailed in [26]. Mosquitoes were allowed to membrane-feed on the blood samples for 20–30 min. Upon feeding, fully engorged blood-fed mosquitoes were collected and used for experimental purposes.

### 2.5. Experimental Design

In this study, the effect of mixed blood meals on the quality of human DNA profiling over the complete digestion profile post-feeding was investigated. Six groups of 6-day-old mosquitoes (100 mosquitoes in each group) were starved for 12 h prior to membrane feeding on human or mixed human-mouse blood. Groups 1 and 2 (G1 and G2) were allowed to feed on the blood of a human male and a human female, respectively. Group 3 (G3) was allowed to feed on equally mixed blood (1:1, *v:v*) of the human male and female. Groups 4 and 5 (G4 and G5) were allowed to feed on equally mixed blood (1:1, *v:v*) of the human female and mouse, and the human male and mouse, respectively. Group 6 (G6) was allowed to feed on equally mixed blood of the human male, human female, and mouse (1:1:1, *v:v:v*).

Immediately after blood feeding, 100 fully engorged female mosquitoes from each mosquito group were randomly collected, assigned to small cages (10 × 10 × 10 cm^3^), and allowed access to 10% glucose solution ad libitum until used for dissection. In each group, five fully engorged females (*n* = 5) were randomly collected at each of the experimental time intervals (0, 2, 4, 6, 8, 10, 12, 24, 36, 48, and 72 h post-feeding) to cover the complete digestion period [18,27]. Blood-fed mosquitoes were stored at −80 °C until used for dissection. Mosquitoes were dissected in cold *Aedes* physiological saline (150 mmolL^−1^ NaCl, 4 mmolL^−1^ KCl, 1 mmolL^−1^ CaCl_2_, 0.1 mmolL^−1^ NaHCO_3_, 0.6 mmolL^−1^ MgCl_2_ buffered with 25 mmolL^−1^ Hepes) according to [28] under a stereo dissection microscope (Optech^®^, Exacta Optech, München, Germany) for extracting their whole abdomens. In parallel with each relevant experiment, unfed mosquito abdomens and 6 µL of human or mouse blood samples, which is equal to a mosquito blood meal size [29,30], were used as controls. Control samples and mosquito blood meals were used for DNA extraction, as detailed below.

### 2.6. DNA Extraction and Quanitation

Human DNA was extracted from the abdomens of control and blood-feed mosquitoes, human blood, and mouse blood samples using the QIAamp DNA micro kit (Qiagen Co., Manchester, UK), following the manufacturer’s instructions and as detailed in [13]. DNA concentration was measured using the Quantifiler^®^ Duo DNA Quantification Kit (Applied Biosystems, Foster City, CA, USA), following the manufacturer’s instructions. Extracted DNAs were used for STR profiling from control and experimental blood samples, as detailed below. 

### 2.7. STRs Profiling 

A polymerase chain reaction was performed using the PowerPlex^®^ Fusion System (Promega, Madison, WI, USA) according to [31] following the instruction manual. This system co-amplifies 23 STR loci (D3S1358, D1S1656, D2S441, D10S1248, D13S317, Penta E, D16S539, D18S51, D2S1338, CSF1PO, Penta D, TH01, vWA, D21S11, D7S820, D5S818, TPOX, DYS391, D8S1179, D12S391, D19S433, FGA, D22S1045) and the Amelogenin locus for sex determination [32]. PCR amplification was carried out using the Applied Biosystems Veriti™ Thermal Cycler (Thermofisher Scientific, Waltham, MA, USA). The PCR products (1 µL) were separated by capillary electrophoresis in an ABI 3500 Genetic Analyzer (Thermo Fisher Scientific Company, Carlsbad, CA, USA) with reference to the BTO size standard (Qiagen, Manchester, UK) in a total of 12 µL master mix consisting of the BTO size standard and Hi-Di formamide (Thermo Fisher Scientific, Inc., Waltham, MA, USA). The analytical threshold for the PowerPlex^®^ Fusion analysis in GeneMapper ID-X was set at 175 RFU. However, the allele call for some peaks was cleaned manually.

### 2.8. Data Analysis

Statistical analysis was performed for DNA concentrations using Prism software (v6.0; GraphPad), while data from STR profiling were analyzed using Gene Mapper^®^ ID v3.2 software (Applied Biosystems, Foster City, CA, USA).

## 3. Results

### 3.1. DNA Quantification

DNA was successfully recovered from mosquito blood meals until 36 h post-feeding. The concentration of each DNA sample was quantified in triplicate (*n* = 3), and the mean values were calculated. As shown in Figure 1, the correlation between concentrations of DNA extracted from the blood meals of the different experimental mosquito groups at each of the experimental time intervals post-feeding until it became weakly detectable at 48 h and undetectable at 72 h post-feeding. These data revealed that DNA concentrations gradually decreased over time post-feeding in all experimental groups. The data in Table 1 represent the concentration and Ct values of DNA in all experimental groups at each time point post-blood meal. The average DNA concentrations after immediate extraction (0 h post-feeding) were 1.25, 1.30, 1.73, 1.29, 1.25, and 1.04 ng/µL in the blood meals of the 6 experimental groups: G1, G2, G3, G4, G5, and G6, respectively. These concentrations declined to 0.004, 0.007, 0.008, 0.007, 0.002, and 0.001 ng/µL at 48 h post-feeding. The concentrations of DNA extracted from control blood samples were significantly higher (*p* ≥ 0.05; *n* = 3; student’s *t*-Test) (ranging between 144.39 and 288.17 ng/μL) compared to those of DNA extracted from experimental mosquito blood meals (Table 1).

### 3.2. STR Profiling 

Data from STR genotyping analysis showed full STR profiles obtained from all mosquito blood meals at 0–12 h post-feeding in all experimental groups and the control samples. However, after 12 h post-feeding, the frequencies of STR loci in DNA profiles decreased with increasing post-feeding time intervals, regardless of the type of blood used to feed mosquitoes. Moreover, partial STR profiles were obtained from mosquito blood meals 24–48 h post-feeding. Figure 2a–d shows an example of the STR profiles from the mosquito blood meal of the first group (G1) (taken from a human male) at 12, 24, 36, and 48 h post-feeding.

Data in Table 2, Table 3, Table 4, Table 5, Table 6 and Table 7 represent the STR profiles for the entire set for analyses of all the experimental groups (G1–G6). Complete profiles for the 24 STR loci were obtained from all groups through (0–12 h) post-blood feeding. A complete STR profile was obtained at 24 h post-feeding. In comparison, between the types of blood meals offered to mosquitoes, full STR profiles were obtained up to 24 h from mosquitoes fed on human female blood, mixed human male-human female blood, and mixed human male-human female-mouse blood (Table 2, Table 4 and Table 7, respectively). However, only partial STR profiles were obtained after 24 h post-feeding, but this was still informative in most groups, except for mosquito blood meals taken from human males (Table 3). 

Data in Table 4 and Table 5 represent incomplete STR profiles obtained from blood meals after 36 h post-feeding, except those obtained from blood meals taken from human female and mixed human female-human male blood (experimental groups G2 and G3, respectively), with only two missing STR loci (vWA and DYS391). On the other hand, only three STR loci (D18S51, D8S1179, and FGA) were obtained from blood meals of G1 (taken from human male blood), and two incomplete STR loci (D19S433 and FGA) were obtained from blood meals of G5 (taken from mixed human female-mice blood) were detected after 36 h post-feeding, respectively (Table 3 and Table 7). Additionally, five STR loci (D18S51, D8S1179, D12S391, D19S433, and FGA) obtained from blood meals of G4 (taken from mixed human male-mouse blood) and six STR loci (Amelogenin, D3S1358, D1S1656, D2S441, D10S1248, and D13S317) obtained from blood meals of G6 (taken from mixed human male-human female-mouse blood) were detected after 36 h post-feeding, as shown in Table 6 and Table 7.

At 48 h post-feeding and onward, no STR profiles were obtained from blood meals of the experimental groups G1 (taken from human male blood), G3 (taken from mixed human male-human female blood), and G5 (taken from mixed human male-mouse blood). However, partial STR profiles were obtained from blood meals of groups G2 (taken from human female blood), G4 (taken from mixed human female-mouse blood), and G6 (taken from mixed human male-human female-mouse blood). Blood meals from G2 yielded a partial STR profile with only two missing STR loci (CSF1PO and DYS391) (Table 4). While groups G4 and G6 yielded five STR loci (D18S51, D8S1179, D12S391, D19S433, and FGA) and only one STR locus (Ahmelogenin), respectively (Table 6 and Table 7). These data may indicate that it is possible to use mosquito blood meals to identify suspects until 36 h post-feeding. It is also possible to identify suspects until the same period of post-feeding, even when mixed with non-human blood.

## 4. Discussion

Insects collected at the crime scene can serve as a source of DNA information in forensic casework [33]. This trace DNA can be obtained from a human blood meal ingested by hematophagous insects found at or nearby the crime scene, like lice [34], bed bugs [35], and mosquitoes [3,14]. These insects need blood meals mainly for vitellogenesis (egg production) [6]. In mosquitoes, the blood meal size and the time elapsed for its complete digestion vary among species and are influenced by several factors. These factors include age, mating behavior, gonotrophic cycle, ambient temperature, and blood meal source [5,6,8,18,29,30]. The *Cx. pipiens* L. mosquito targeted in the current study takes about 60 to 70 hours to completely digest the blood meal [18]. The current study investigated the capability of human DNA profiling over a 72 h period post-blood meal.

There are four important characteristics that strongly support the practical use of mosquitoes as a forensic substrate at a crime scene: (a) there are over 3600 species of mosquitoes, and some are commonly found worldwide [36]; (b) there are variations in mosquito behavior, including differences in flight, feeding time, and host preference. As a result, these behavioral variations increase the likelihood of use in forensic applications [3]; (c) The blood-fed mosquitoes often rest at or near the location of their host until they have digested enough of the blood to be able to fly [16,37]. This is advantageous from the forensic point of view, as blood-fed mosquitoes are more likely to rest at or near a crime scene for digestion; (d) Mosquitoes can feed on multiple blood sources [38], which provides an opportunity to detect and interpret the profiles of multiple hosts. Using a mosquito, if found at a crime scene and a mixed profile is obtained from its blood meal, to pursue more than one person engaged in the crime or present nearby. Several studies have successfully used blood-fed mosquitoes found at a crime scene to obtain forensically valuable human DNA profiles, which are then used to identify potential suspects in forensic cases [11,14].

The quantification of DNA extracted from crime scene trace samples is an important factor in forensic DNA analysis because it determines whether the DNA data is of sufficient quantity and quality to produce a successful STR profile [15]. It is sometimes challenging to conduct forensic analysis of human DNA from trace samples since it can be affected by some factors like degradation, insufficient amounts, and low quality and, subsequently, can limit the analysis process [9,11,12,18]. As the Ct value increases, the possibility of sample contamination or DNA degradation could be possible. The reliability of detecting reasonable copy numbers of DNA increases if the Ct value is lower than 40. In this study, Ct values indicate that DNA was detectable and amplifiable [39]. The current study included 24 variable-sized STR loci, including amelogenin, to allow a broader analysis of the human DNA samples extracted from the mosquito blood meal and also because these loci include the CODIS core loci and European Standard Set (ESS) loci [3,19,40]. Some DNA samples in the current study had insufficient quantities, which could be due to two possible reasons: the first is that mosquitoes were fed on mixed blood samples treated with heparin as anticoagulants instead of being exposed directly to the human body [12]. The second is the digestion/nuclease activity in the mosquito’s gut that could cause DNA degradation and, consequently, a reduction in DNA quantity [41]. Further studies might be conducted to compare the efficiency of different DNA extraction kits in obtaining sufficient DNA quantities that have been obtained by using the QIAamp DNA micro kit (Qiagen Co., Ltd., Manchester, UK) alone. This could increase the possibility of overcoming the abovementioned limitations and produce more allele frequencies at 48 h post-feeding.

This study reported complete STR profiles ranging from 0.003 to 1.74 (ng/μL) and ≈30 to 1740 (pg/µL) in the human DNA extracted from mosquito blood meals. Previous studies on the autosomal STR loci of the U.S. population data produced full STR profiles even from fewer DNA quantities [42]. Moreover, 15 forensically valid genetic loci were successfully obtained from a 28 pg DNA concentration [9]. In the current study, STR profiles were obtained up to 36 h post-feeding, after which they became poorly detectable at 48 h post-feeding. This could be attributed to DNA degradation in the mosquito’s midgut by the digestive enzymes [15,41]. Several previous studies reported successful amplification of complete STR profiles from human DNA extracted from mosquitoes, but with variations in the maximum post-feeding interval that could still yield full profiles. In some cases, full STR profiles were obtained up to 24 h post-feeding of *Ae. aegypti* on mixed blood of four people [12], while others obtained full STR profiles up to 48 h post-feeding of other mosquito types [9,16,43] and 72 h post-feeding [35]. These variations could be attributed to different factors, such as the mosquito species, the volume of ingested blood, the inhibitors found in the mosquito’s midgut, the digestion status, and other factors related to the host (human), such as sex and environmental conditions [6]. Moreover, the results confirmed a higher reduction in the allele frequencies in the human male DNA samples than that in the human female. Similar findings were reported in previous studies [3,44]. DNA samples from mixed blood meals from human males and/or females with mouse blood showed less number of alleles than non-mixed human blood samples and mixed human male-human female blood meals. The study also revealed that the interval time pertaining to digestion status on the mosquito’s blood meal had a strong impact on both the DNA quantity and allele frequency. Further, the study also revealed an extensive reduction in the frequency of alleles at 36 to 48 h post-feeding. Evidence for similar observations has been provided by previous studies [15]. 

The number of STR loci reported in the current study was higher in the human female DNA sample than that of the male. This difference could be attributed to our personal observation that mosquitoes prefer female blood during feeding, as discussed earlier. However, a previous study reported similar mosquito feeding (biting) patterns in both men and women [9]. Other previous studies revealed that *Anopheles stephensi*, *Aedes aegypti*, and *Anopheles gambiae* preferred men’s blood because of odor and hormonal factors [6,45,46]. In this context, many factors affect the feeding preferences of female mosquitoes, such as blood viscosity, temperatures, pH, and salt contents (blood osmosis) [6]. Mosquitoes are twice as likely to be attracted to blood with high CO_2_ levels and a higher body temperature and they can detect pregnant females [6,46]. Therefore, in this study, it was necessary to fix the temperature of the experimental membrane feeder to 37 °C and the blood group (O^+^) for both male and female human volunteers’ blood during the experiments. In addition, human female blood differs from that of the male in viscosity, salt contents, and hematocrit, which could contribute to the variations in mosquito feeding preferences and digestion rate [45,46]. These differences may explain the differences in DNA quantities and, consequently, the STR profiling reported in the study, as the blood meal from human female blood yielded a partial profile up to 48 h post-feeding compared to 36 h for that of human male blood. 

## 5. Conclusions

In conclusion, this study provides evidence supporting *Cx. pipiens* L. mosquitoes as a possible candidate for providing forensically valid human DNA as hard evidence that may aid the crime investigators in identifying persons and decoding ambiguous crimes. Mixed blood meals taken from humans and animals may have contributed to maximizing DNA degradation and consequently affected the STR identification. Thus, despite the potentially degraded DNA revealed in this study at 48 h post-feeding, it was possible to record profiles with a forensically valuable number of STR loci up to 36 h post-feeding in the examined blood meals. It is preferable for technicians to collect as many blood-fed mosquitoes as possible from a crime scene in order to maximize the recovery of human-specific DNA from their blood meals. This would increase the possibility of producing complete DNA profiles, aid in the identification of a potential offender, and/or exclude a suspect. 

## Figures and Tables

**Figure 1 insects-14-00467-f001:**
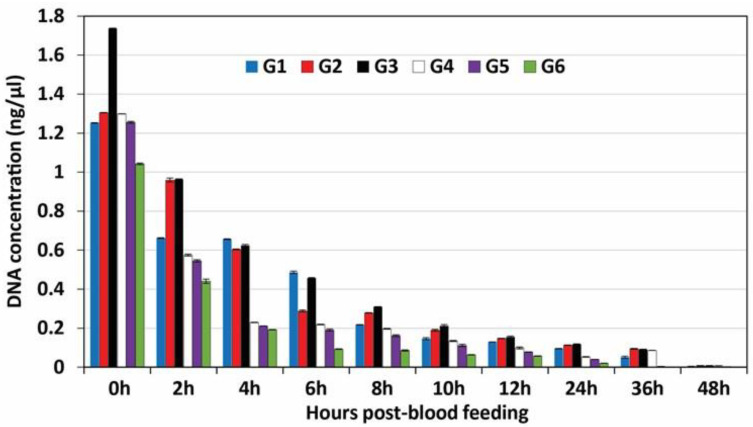
DNA concentration (ng/μL) extracted from the blood meals of the six experimental mosquito groups at different time intervals post-feeding.

**Figure 2 insects-14-00467-f002:**
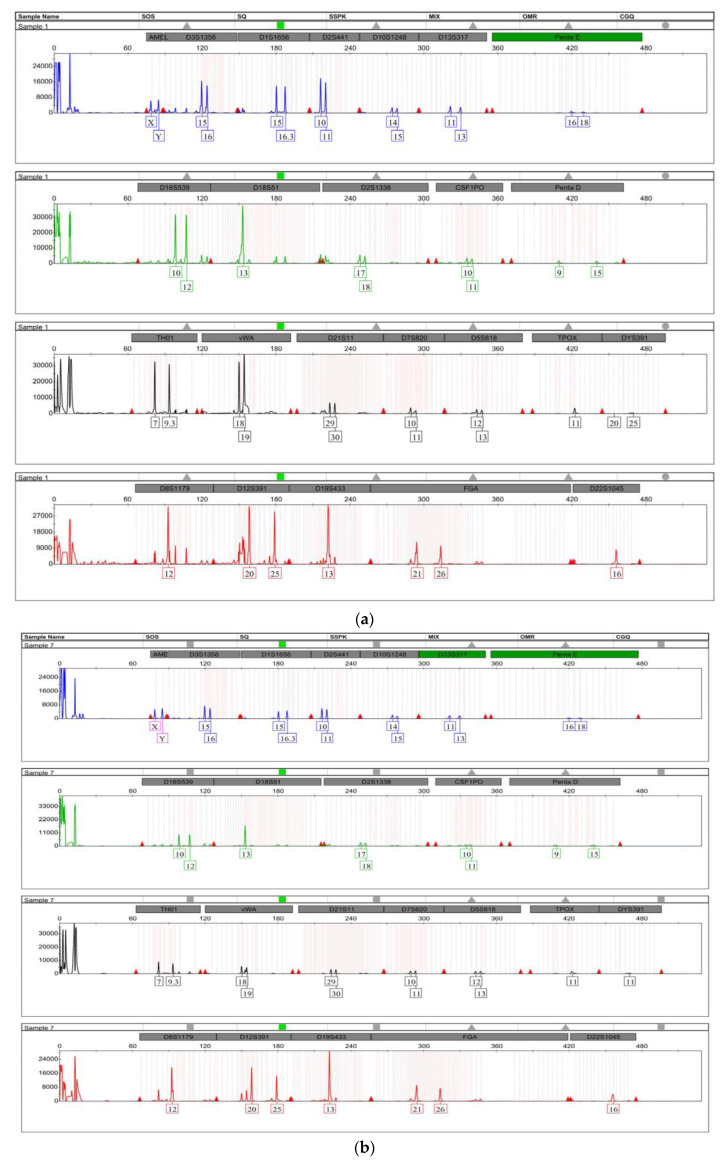
(**a**). STR profile from the blood meal of “G1” (taken from human male blood) at 12 h post-feeding. (**b**). STR profile from the blood meal of “G1” (taken from human male blood) at 24 h post-feeding). (**c**). STR profile from the blood meal of “G1” (taken from human male blood) at 36 h post-feeding. (**d**). STR profile from the blood meal of “G1” (taken from human male blood) at 48 h post-feeding.

**Table 1 insects-14-00467-t001:** Mean concentrations (ng/μL) and cycle threshold values of the DNA extracted from the blood meals of the six experimental mosquito groups at different time intervals post-feeding compared to the control samples.

Hours Post-Feeding	Mosquito Groups (Based on the Source of Blood Meals)											
	G1		G2		G3		G4		G5		G6	
	C	C_T_	C	C_T_	C	C_T_	C	C_T_	C	C_T_	C	C_T_
0	1.252 ± 0.001	29.91	1.305 ± 0.001	29.03	1.736 ± 0.001	28.08	1.298 ± 0.0005	29.69	1.255 ± 0.005	29.87	1.042 ± 0.004	31.36
2	0.661 ± 0.001	32.28	0.959 ± 0.01	31.98	0.963 ± 0.002	31.96	0.574 ± 0.005	32.34	0.545 ± 0.006	32.38	0.44 ± 0.011	32.51
4	0.656 ± 0.002	32.29	0.604 ± 0.002	32.28	0.623 ± 0.006	32.28	0.229 ± 0.001	33.77	0.211 ± 0.0005	33.79	0.192 ± 0.0005	33.85
6	0.486 ± 0.006	32.31	0.288 ± 0.005	33.46	0.457 ± 0.002	32.49	0.218 ± 0.001	33.76	0.19 ± 0.005	33.83	0.092 ± 0.001	35.03
8	0.217 ± 0.001	33.78	0.278 ± 0.001	33.49	0.309 ± 0.001	33.43	0.197 ± 0.002	33.84	0.161 ± 0.005	33.92	0.085 ± 0.003	35.24
10	0.145 ± 0.005	33.98	0.189 ± 0.005	33.85	0.211 ± 0.006	33.79	0.134 ± 0.003	33.99	0.111 ± 0.006	34.02	0.063 ± 0.001	35.12
12	0.129 ± 0.0005	33.10	0.147 ± 0.001	33.99	0.154 ± 0.005	33.97	0.098 ± 0.0005	35.08	0.077 ± 0.001	35.11	0.056 ± 0.001	35.13
24	0.094 ± 0.001	35.10	0.112 ± 0.001	34.02	0.118 ± 0.001	33.87	0.052 ± 0.001	35.15	0.039 ± 0.0005	35.25	0.019 ± 0.0005	35.45
36	0.05 ± 0.005	35.19	0.094 ± 0.002	35.11	0.091 ± 0.0005	35.14	0.086 ± 0.0005	35.14	0.003 ± 0.0005	38.88	0.001 ± 0.00	44.18
48	0.004 ± 0.0005	38.86	0.007 ± 0.001	38.84	0.008 ± 0.0005	38.83	0.007 ± 0.0005	38.84	0.002 ± 0.00	39.16	0.001 ± 0.00	44.21
Control	144.4 * ± 0.34	22.1	241.8 * ± 1.03	21.8	288.2 * ± 0.69	21.3	237.99 * ± 1.14	21.86	0.001 * ± 0.00	44.21	145.1 * ± 0.6	22.14

C: Concentration; C_T_: Cycle threshold; G1: Group 1; G2: Group 2; G3: Group 3; G4: Group 4; G5: Group 5, and G6: Group 6; * significantly higher compared to that of experimental group (*p* ≤ 0.05, Students *t*-Test, *n* = 3).

**Table 2 insects-14-00467-t002:** DNA genotyping for 23 STR loci and Amelogenin obtained from mosquito blood meals taken from human male blood (G1) at different time intervals post-feeding.

STR Loci	Hours Post-Blood Feeding										
	0 h	2 h	4 h	6 h	8 h	10 h	12 h	24 h	36 h	48 h	Control
Amelogenin	XY	XY	XY	XY	XY	XY	XY	N/A	N/A	N/A	XY
D3S1358	15,16	15,16	15,16	15,16	15,16	15,16	15,16	15,16	N/A	N/A	15,16
D1S1656	15,16.3	15,16.3	15,16.3	15,16.3	15,16.3	15,16.3	15,16.3	15,16.3	N/A	N/A	15,16.3
D2S441	10,11	10,11	10,11	10,11	10,11	10,11	10,11	10,11	N/A	N/A	10,11
D10S1248	14,15	14,15	14,15	14,15	14,15	14,15	14,15	N/A	N/A	N/A	14,15
D13S317	11,13	11,13	11,13	11,13	11,13	11,13	11,13	N/A	N/A	N/A	11,13
Penta E	16,18	16,18	16,18	16,18	16,18	16,18	16,18	N/A	N/A	N/A	16,18
D16S539	10,12	10,12	10,12	10,12	10,12	10,12	10,12	10,12	N/A	N/A	10,12
D18S51	13,13	13,13	13,13	13,13	13,13	13,13	13,13	13,13	13,13	N/A	13,13
D2S1338	17,18	17,18	17,18	17,18	17,18	17,18	17,18	N/A	N/A	N/A	17,18
CSF1PO	10,11	10,11	10,11	10,11	10,11	10,11	10,11	N/A	N/A	N/A	10,11
Penta D	9,15	9,15	9,15	9,15	9,15	9,15	9,15	N/A	N/A	N/A	9,15
TH01	7,9.3	7,9.3	7,9.3	7,9.3	7,9.3	7,9.3	7,9.3	7,9.3	N/A	N/A	7,9.3
vWA	18,19	18,19	18,19	18,19	18,19	18,19	18,19	18,19	N/A	N/A	18,19
D21S11	29,30	29,30	29,30	29,30	29,30	29,30	29,30	N/A	N/A	N/A	29,30
D7S820	10,11	10,11	10,11	10,11	10,11	10,11	10,11	N/A	N/A	N/A	10,11
D5S818	12,13	12,13	12,13	12,13	12,13	12,13	12,13	N/A	N/A	N/A	12,13
TPOX	9,11	9,11	9,11	9,11	9,11	9,11	9,11	N/A	N/A	N/A	11,11
DYS391	11,11	11,11	11,11	11,11	11,11	11,11	11,11	11,11	N/A	N/A	11,11
D8S1179	12,12	12,12	12,12	12,12	12,12	12,12	12,12	12,12	12,12	N/A	12,12
D12S391	20,25	20,25	20,25	20,25	20,25	20,25	20,25	20,25	N/A	N/A	20,25
D19S433	13,13	13,13	13,13	13,13	13,13	13,13	13,13	13,13	N/A	N/A	13,13
FGA	21,26	21,26	21,26	21,26	21,26	21,26	21,26	N/A	21,26	N/A	21,26
D22S1045	15,16	15,16	15,16	15,16	15,16	15,16	15,16	N/A	N/A	N/A	15,16

N/A: No results.

**Table 3 insects-14-00467-t003:** DNA genotyping for 22 STR loci and amelogenin obtained from mosquito blood meals taken from human female blood (G2) at different time intervals post-feeding.

STR Loci	Hours Post-Blood Feeding										
	0	2	4	6	8	10	12	24	36	48	Control
Amelogenin	XX	XX	XX	XX	XX	XX	XX	XX	XX	XX	XX
D3S1358	17,17	17,17	17,17	17,17	17,17	17,17	17,17	17,17	17,17	17,17	17,17
D1S1656	15.3,16.3	15.3,16.3	15.3,16.3	15.3,16.3	15.3,16.3	15.3,16.3	15.3,16.3	15.3,16.3	15.3,16.3	15.3,16.3	15.3,16.3
D2S441	10,14	10,14	10,14	10,14	10,14	10,14	10,14	10,14	10,14	10,14	10,14
D10S1248	13,14	13,14	13,14	13,14	13,14	13,14	13,14	13,14	13,14	13,14	13,14
D13S317	12,14	12,14	12,14	12,14	12,14	12,14	12,14	12,14	12,14	12,14	12,14
Penta E	16,18	16,18	16,18	16,18	16,18	16,18	16,18	16,18	16,18	16,18	16,18
D16S539	9,10	9,10	9,10	9,10	9,10	9,10	9,10	9,10	9,10	9,10	9,10
D18S51	13,16	13,16	13,16	13,16	13,16	13,16	13,16	13,16	13,16	13,16	13,16
D2S1338	17,18	17,18	17,18	17,18	17,18	17,18	17,18	17,18	17,18	17,18	17,18
CSF1PO	10,11	10,11	10,11	10,11	10,11	10,11	10,11	10,11	10,11	N/A	10,11
Penta D	9,16	9,16	9,16	9,16	9,16	9,16	9,16	9,16	9,16	9,16	9,16
TH01	6,7	6,7	6,7	6,7	6,7	6,7	6,7	6,7	6,7	6,7	6,7
vWA	18,19	18,19	18,19	18,19	18,19	18,19	18,19	18,19	18,19	18,19	18,19
D21S11	30,32.2	30,32.2	30,32.2	30,32.2	30,32.2	30,32.2	30,32.2	30,32.2	30,32.2	30,32.2	30,32.2
D7S820	11,11	11,11	11,11	11,11	11,11	11,11	11,11	11,11	11,11	11,11	11,11
D5S818	12,14	12,14	12,14	12,14	12,14	12,14	12,14	12,14	12,14	12,14	12,14
TPOX	9,11	9,11	9,11	9,11	9,11	9,11	9,11	9,11	9,11	9,11	9,11
D8S1179	14,15	14,15	14,15	14,15	14,15	14,15	14,15	14,15	14,15	14,15	14,15
D12S391	23,23	23,23	23,23	23,23	23,23	23,23	23,23	23,23	23,23	23,23	23,23
D19S433	12,13	12,13	12,13	12,13	12,13	12,13	12,13	12,13	12,13	12,13	12,13
FGA	21,23	21,23	21,23	21,23	21,23	21,23	21,23	21,23	21,23	21,23	21,23
D22S1045	15,16	15,16	15,16	15,16	15,16	15,16	15,16	15,16	15,16	15,16	15,16

N/A: No results.

**Table 4 insects-14-00467-t004:** DNA genotyping for 23 STR loci and amelogenin obtained from mosquito blood meals taken from mixed human male-human female blood (G3) at different time intervals post-feeding.

STR Loci	Hours Post-Blood Feeding										
	0 h	2 h	4 h	6 h	8 h	10 h	12 h	24 h	36 h	48 h	Control
Amelogenin	XX,Y	XX,Y	XXY	XX,Y	XX,Y	XX,Y	XX,Y	XX,Y	XX,Y	N/A	XX,Y
D3S1358	15,16,17	15,16,17	15,16,17	15,16,17	15,16,17	15,16,17	15,16,17	15,16,17	15,16,17	N/A	15,16,17
D1S1656	15,15.3,16.3	15,15.3,16.3	15,15.3,16.3	15,15.3,16.3	15,15.3,16.3	15,15.3,16.3	15,15.3,16.3	15,15.3,16.3	15,15.3, 16.3	N/A	15,15.3,16.3
D2S441	10,11,14	10,11,14	10,11,14	10,11,14	10,11,14	10,11,14	10,11,14	10,11,14	10,11,14	N/A	10,11,14
D10S1248	13,14,15	13,14,15	13,14,15	13,14,15	13,14,15	13,14,15	13,14,15	13,14,15	13,14,15	N/A	13,14,15
D13S317	11,12,13,14	11,12,13,14	11,12,13,14	11,12,13,14	11,12,13,14	11,12,13,14	11,12,13,14	11,12,13,14	11,12,13,14	N/A	11,12,13,14
Penta E	16,18	16,18	16,18	16,18	16,18	16,18	16,18	16,18	16,18	N/A	16,18
D16S539	9,10,12	9,10,12	9,10,12	9,10,12	9,10,12	9,10,12	9,10,12	9,10,12	9,10,12	N/A	9,10,12
D18S51	13,16	13,16	13,16	13,16	13,16	13,16	13,16	13,16	13,16	N/A	13,16
D2S1338	17,18	17,18	17,18	17,18	17,18	17,18	17,18	17,18	17,18	N/A	17,18
CSF1PO	10,11	10,11	10,11	10,11	10,11	10,11	10,11	10,11	10,11	N/A	10,11
Penta D	9,15,16	9,15,16	9,15,16	9,15,16	9,15,16	9,15,16	9,15,16	9,15,16	9,15,16	N/A	9,15,16
TH01	6,7,9.3	6,7,9.3	6,7,9.3	6,7,9.3	6,7,9.3	6,7,9.3	6,7,9.3	6,7,9.3	9,15,16	N/A	6,7,9.3
vWA	18,19	18,19	18,19	18,19	18,19	18,19	18,19	18,19	N/A	N/A	18,19
D21S11	29,30,32.2	29,30,32.2	29,30,32.2	29,30,32.2	29,30,32.2	29,30,32.2	29,30,32.2	29,30,32.2	18,19	N/A	29,30,32.2
D7S820	10,11	10,11	10,11	10,11	10,11	10,11	10,11	10,11	10,11	N/A	10,11
D5S818	12,13,14	12,13,14	12,13,14	12,13,14	12,13,14	12,13,14	12,13,14	12,13,14	12,13,14	N/A	12,13,14
TPOX	9,11	9,11	9,11	9,11	9,11	9,11	9,11	9,11	9,11	N/A	9,11
DYS391	7,11	7,11	7,11	7,11	7,11	7,11	7,11	7,11	7, N/A	N/A	7,11
D8S1179	12,14,15	12,14,15	12,14,15	12,14,15	12,14,15	12,14,15	12,14,15	12,14,15	12,14,15	N/A	12,14,15
D12S391	20,23,25	20,23,25	20,23,25	20,23,25	20,23,25	20,23,25	20,23,25	20,23,25	20,23,25	N/A	20,23,25
D19S433	12,13	12,13	12,13	12,13	12,13	12,13	12,13	12,13	12,13	N/A	12,13
FGA	21,23,26	21,23,26	21,23,26	21,23,26	21,23,26	21,23,26	21,23,26	21,23,26	21,23,26	N/A	21,23,26
D22S1045	15,16	15,16	15,16	15,16	15,16	15,16	15,16	15,16	15,16	N/A	15,16

N/A: No results.

**Table 5 insects-14-00467-t005:** DNA genotyping for 23 STR loci and amelogenin obtained from mosquito blood meals taken from mixed male-mouse blood (G4) at different time intervals post-feeding.

STR Loci	Hours Post-Blood Feeding										
	0 h	2 h	4 h	6 h	8 h	10 h	12 h	24 h	36 h	48 h	Control
Amelogenin	XY	XY	XY	XY	XY	XY	XY	XY	N/A	N/A	XY
D3S1358	15,16	15,16	15,16	15,16	15,16	15,16	15,16	15,16	N/A	N/A	15,16
D1S1656	15,16.3	15,16.3	15,16.3	15,16.3	15,16.3	15,16.3	15,16.3	15,16.3	N/A	N/A	15,16.3
D2S441	10,11	10,11	10,11	10,11	10,11	10,11	10,11	10,11	N/A	N/A	10,11
D10S1248	14,15	14,15	14,15	14,15	14,15	14,15	14,15	14,15	N/A	N/A	14,15
D13S317	11,13	11,13	11,13	11,13	11,13	11,13	11,13	11,13	N/A	N/A	11,13
Penta E	16,18	16,18	16,18	16,18	16,18	16,18	16,18	16,18	N/A	N/A	16,18
D16S539	10,12	10,12	10,12	10,12	10,12	10,12	10,12	10,12	N/A	N/A	10,12
D18S51	13,13	13,13	13,13	13,13	13,13	13,13	13,13	13,13	13,13	13,13	13,13
D2S1338	17,18	17,18	17,18	17,18	17,18	17,18	17,18	17,18	N/A	N/A	17,18
CSF1PO	10,11	10,11	10,11	10,11	10,11	10,11	10,11	10,11	N/A	N/A	10,11
Penta D	9,15	9,15	9,15	9,15	9,15	9,15	9,15	9,15	N/A	N/A	9,15
TH01	7,9.3	7,9.3	7,9.3	7,9.3	7,9.3	7,9.3	7,9.3	7,9.3	N/A	N/A	7,9.3
vWA	18,19	18,19	18,19	18,19	18,19	18,19	18,19	18,19	N/A	N/A	18,19
D21S11	29,30	29,30	29,30	29,30	29,30	29,30	29,30	29,30	N/A	N/A	29,30
D7S820	10,11	10,11	10,11	10,11	10,11	10,11	10,11	10,11	N/A	N/A	10,11
D5S818	12,13	12,13	12,13	12,13	12,13	12,13	12,13	12,13	N/A	N/A	12,13
TPOX	9,11	9,11	9,11	9,11	9,11	9,11	9,11	9,11	N/A	N/A	11,11
DYS391	11,11	11,11	11,11	11,11	11,11	11,11	11,11	11,11	N/A	N/A	11,11
D8S1179	12,12	12,12	12,12	12,12	12,12	12,12	12,12	12,12	12,12	12,12	12,12
D12S391	20,25	20,25	20,25	20,25	20,25	20,25	20,25	20,25	20,25	20,25	20,25
D19S433	13,13	13,13	13,13	13,13	13,13	13,13	13,13	13,13	13,13	13,13	13,13
FGA	21,26	21,26	21,26	21,26	21,26	21,26	21,26	21,26	21,26	21,26	21,26
D22S1045	15,16	15,16	15,16	15,16	15,16	15,16	15,16	15,16	N/A	N/A	16,16

N/A: No results.

**Table 6 insects-14-00467-t006:** DNA genotyping for 22 STR loci and amelogenin obtained from mosquito blood meals taken from mixed human female-mouse blood (G5) at different time intervals post-feeding.

STR Loci	Hours Post-Blood Feeding										
	0	2	4	6	8	10	12	24	36	48	Control
Amelogenin	XX	XX	XX	XX	XX	XX	XX	XX	N/A	N/A	XX
D3S1358	17,17	17,17	17,17	17,17	17,17	17,17	17,17	17,17	N/A	N/A	17,17
D1S1656	15.3,16.3	15.3,16.3	15.3,16.3	15.3,16.3	15.3,16.3	15.3,16.3	15.3,16.3	15.3,16.3	N/A	N/A	15.3,16.3
D2S441	10,14	10,14	10,14	10,14	10,14	10,14	10,14	10,14	N/A	N/A	10,14
D10S1248	13,14	13,14	13,14	13,14	13,14	13,14	13,14	13,14	N/A	N/A	13,14
D13S317	12,14	12,14	12,14	12,14	12,14	12,14	12,14	12,14	N/A	N/A	12,14
Penta E	16,18	16,18	16,18	16,18	16,18	16,18	16,18	N/A	N/A	N/A	16,18
D16S539	9,10	9,10	9,10	9,10	9,10	9,10	9,10	9,10	N/A	N/A	9,10
D18S51	13,16	13,16	13,16	13,16	13,16	13,16	13,16	13,16	N/A	N/A	13,16
D2S1338	17,18	17,18	17,18	17,18	17,18	17,18	17,18	17,18	N/A	N/A	17,18
CSF1PO	10,11	10,11	10,11	10,11	10,11	10,11	10,11	10,11	N/A	N/A	10,11
Penta D	9,16	9,16	9,16	9,16	9,16	9,16	9,16	9, N/A	N/A	N/A	9,16
TH01	6,7	6,7	6,7	6,7	6,7	6,7	6,7	6,7	N/A	N/A	6,7
vWA	18,19	18,19	18,19	18,19	18,19	18,19	18,19	18,19	N/A	N/A	18,19
D21S11	30,32.2	30,32.2	30,32.2	30,32.2	30,32.2	30,32.2	30,32.2	30,32.2	N/A	N/A	30,32.2
D7S820	11,11	11,11	11,11	11,11	11,11	11,11	11,11	11,11	N/A	N/A	11,11
D5S818	12,14	12,14	12,14	12,14	12,14	12,14	12,14	12,14	N/A	N/A	12,14
TPOX	9,11	9,11	9,11	9,11	9,11	9,11	9,11	9,11	N/A	N/A	9,11
D8S1179	14,15	14,15	14,15	14,15	14,15	14,15	14,15	14,15	N/A	N/A	14,15
D12S391	23,23	23,23	23,23	23,23	23,23	23,23	23,23	23,23	N/A	N/A	23,23
D19S433	12,13	12,13	12,13	12,13	12,13	12,13	12,13	12,13	12, N/A	N/A	12,13
FGA	21,23	21,23	21,23	21,23	21,23	21,23	21,23	21,23	23, N/A	N/A	21,23
D22S1045	15,16	15,16	15,16	15,16	15,16	15,16	15,16	N/A	N/A	N/A	15,16

N/A: No results.

**Table 7 insects-14-00467-t007:** DNA genotyping for 23 STR loci and amelogenin obtained from mosquito blood meals taken from mixed human male-human female-mouse blood (G6) at different time intervals post-feeding.

STR Loci	Hours Post-Blood Feeding										
	0	2	4	6	8	10	12	24	36	48	Control
Amelogenin	XX,Y	XX,Y	XX,Y	XX,Y	XX,Y	XX,Y	XX,Y	XX,Y	XX,Y	XX,Y	XX,Y
D3S1358	15,16,17	15,16,17	15,16,17	15,16,17	15,16,17	15,16,17	15,16,17	15,16,17	15,16,17	N/A	15,16,17
D1S1656	15,15.3,16.3	15,15.3,16.3	15,15.3,16.3	15,15.3,16.3	15,15.3,16.3	15,15.3,16.3	15,15.3,16.3	15,15.3,16.3	15,15,16.3	N/A	15,15.3,16.3
D2S441	10,11,14	10,11,14	10,11,14	10,11,14	10,11,14	10,11,14	10,11,14	10,11,14	10,11,14	N/A	10,11,14
D10S1248	13,14,15	13,14,15	13,14,15	13,14,15	13,14,15	13,14,15	13,14,15	13,14,15	13,14,15	N/A	13,14,15
D13S317	11,12,13,14	11,12,13, 14	11,12,13, 14	11,12,13, 14	11,12,13, 14	11,12,13, 14	11,12,13, 14	11,12,13, 14	11,12,13,14	N/A	11,12,13, 14
Penta E	16,18	16,18	16,18	16,18	16,18	16,18	16,18	16,18	N/A	N/A	16,18
D16S539	9,10,12	9,10,12	9,10,12	9,10,12	9,10,12	9,10,12	9,10,12	9,10,12	N/A	N/A	9,10,12
D18S51	13,16	13,16	13,16	13,16	13,16	13,16	13,16	13,16	N/A	N/A	13,16
D2S1338	17,18	17,18	17,18	17,18	17,18	17,18	17,18	17,18	N/A	N/A	17,18
CSF1PO	10,11	10,11	10,11	10,11	10,11	10,11	10,11	10,11	N/A	N/A	10,11
Penta D	9,15,16	9,15,16	9,15,16	9,15,16	9,15,16	9,15,16	9,15,16	9,15, N/A	N/A	N/A	9,15,16
TH01	6,7,9.3	6,7,9.3	6,7,9.3	6,7,9.3	6,7,9.3	6,7,9.3	6,7,9.3	6,7,9.3	N/A	N/A	6,7,9.3
vWA	18,19	18,19	18,19	18,19	18,19	18,19	18,19	18,19	N/A	N/A	18,19
D21S11	29,30,32.2	29,30,32.2	29,30,32.2	29,30,32.2	29,30,32.2	29,30,32.2	29,30,32.2	29,30,32.2	N/A	N/A	29,30,32.2
D7S820	10,11	10,11	10,11	10,11	10,11	10,11	10,11	10,11	N/A	N/A	10,11
D5S818	11,12,13,14	12,13,14	12,13,14	12,13,14	12,13,14	12,13,14	12,13,14	12,13,14	N/A	N/A	12,13,14
TPOX	9,11	9,11	9,11	9,11	9,11	9,11	9,11	9,11	N/A	N/A	9,11
DYS391	7,11	7,11	7,11	7,11	7,11	7,11	7,11	7, N/A	N/A	N/A	7,11
D8S1179	12,14,15	12,14,15	12,14,15	12,14,15	12,14,15	12,14,15	12,14,15	12,14,15	N/A	N/A	12,14,15
D12S391	20,23,25	20,23,25	20,23,25	20,23,25	20,23,25	20,23,25	20,23,25	20,23,25	N/A	N/A	20,23,25
D19S433	12,13	12,13	12,13	12,13	12,13	12,13	12,13	12,13	N/A	N/A	12,13
FGA	21,23,26	21,23,26	21,23,26	21,23,26	21,23,26	21,23,26	21,23,26	21,23,26	N/A	N/A	21,23,26
D22S1045	15,16	15,16	15,16	15,16	15,16	15,16	15,16	15,16	N/A	N/A	15,16

N/A: No results.

## Data Availability

All data pertinent to this work are presented in the paper. Any requests should be directed to the corresponding authors.

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
