# Peer review of "Forensic DNA Analysis of Mixed Mosquito Blood Meals: STR Profiling for Human Identification"

_insects, 2023, doi:10.3390/insects14050467_

Round 1
Reviewer 1 Report
nice (part) replication study with good references
line 53/54: this sentence does not contain information, pls delete
line 55: no, not all studies that you quote observed and documented mosquitos. please check the studies; some are pure laboratory studies
line 61: you mean "from mosquitos", not "on the human DNA"
line 73: what lasts 60 hrs, digestion or degradation? pls be precise.
line 78: why expensive? why time-consuming? these are relatively normal laboratory tasks?
line 82/83: i do not understand the sentence: "thus"? why "thus"? a simple stain also is of forensic value.
line 94 & 149 & 172 & 189: name of author not necessary, number of citation is sufficient
line 94: what do you mean by "briefly"?
line 96 & 110 & 133: ONE mouse? or several mice?
line 100: approximately 200? or did you count?
line 127 & 204 & 477: one 'empty space' too much
line 140: what do you mean by "aspirated"?
line 169: really -80 deg celsius? did you check this?
line 197: "for data comparision"? maybe delete this phrase?
line 203: 36 SPACE h
line 211 to 214: pls round off values
line 238: "extracted from the blood meals" → what exactly do you mean?
line 339: "are interferred" → pls check grammar
line 404: "this trace of DNA" → pls check style (did you mean "trace DNA"?)
line 411, 414, 426, 435, 438, 461, 490, 494: "thus", "in this context", "in this regard", "further", "meanwhile", "however", "therefore" → pls delete 'empty' phrases, they contain no information
line 446: your quote [39] does not refer to mosquito blood, pls mention that
line 450 & 468: why do you refer to [37] here? pls explain in text, i am not sure if this is what you really meant?
line 478: females (not femailes)
line 489: "decoding ambiguous" → pls check language
line 494: "which are suitable for human identification" → pls delete, you stated this already (and do it again afterwards)
line 500: what do you mean with "formal"? pls be more specific
line 504: validation of what? pls be specific
language could be a little more precise; this might be a translation issue but it needs to be checked again pls because the translator probably did sometimes not understand what was meant exactly. a little more "sharp", scientific style would be good.
empty filler words should be removed ("thus", "in this context", "in this regard", "further", "meanwhile", "however", "therefore")
Reviewer 2 Report
Reviewer comments
Manuscript title: Forensic analysis of mixed mosquito blood meals: STR profiling for human identification
The study evaluated the possibility of extracting human STRs from mosquitoes, which would be valuable for forensic investigations to determine whether suspects were present around the scene. The manuscript has several major issues should be addressed, otherwise the manuscript will not be suitable for publication in Insects.
(1) In Table 3 and Table 6, the authors detected YSTR in mosquito blood meals taken from human female blood. This is a serious problem. DYS391 is the STR locus of Y chromosome. Female does not have Y chromosome.
(2) During the PCR process, did the author consider increasing the templates of DNA to improve the amplification efficiency? For example, for samples < 24 h, 1 microliter DNA templates was added to amplification. For sample > 24 h, added 2 microliter templates.
(3) The Powerplex® Fusion System was selected for STR typing in this study. Did the author consider using other kits to verify loci, or using other kits to increase the number of loci detected? The sequence of loci is different in different kits. Loci that do not have peak in the Powerplex® Fusion System may have peak normally in other kits.
Other comments:
Line 26: Please provide the authority name when a species was first mentioned. In addition, the genus name is first written in full and then abbreviated. I found many similar problems in the manuscript. Please check the full manuscript carefully and revise them.
Line 30: "-" should be deleted.
Line 41: Culex pipiens should be in italic.
Line 41: “3” should be superscript.
Line 127: Please check this line and other parts of manuscript carefully for extra “Spaces”.
Line 164: Eleven time points, 5 samples each time point, there should be a total of 55 samples.
Line 456: Could the difference be related to the age of the donors? The age difference between male and female donors in this study was very large.
Figures: The resolution of all images are poor, please provide clearer pictures. The captions of Figures 2b and 2d have no corresponding pictures.
Reviewer 3 Report
For the benefits of the authors, here are my specific comments for consideration:
a) Title
The title appeared to be appropriate with regards to the work described in the manuscript. However, consider the use of forensic DNA analysis instead of using the word forensic analysis which has a broader meaning. STR DNA profiling also should be considered to indicate specific analysis for human identification.
b) Abstract:
The motivation and problem statement to represent the justification for performing the work should be emphasized more by the authors.
· Please consider adding a wider scope of identification not only focusing on suspects alone (e.g., victim)
· The authors used the term “…STR profiles”, consider using DNA profiles instead.
· Please check for grammatical errors in the paragraph.
c) Introduction:
Generally, the introduction does contain information that is essential for describing the background of the study. However, please consider to re-arrange your introduction section based on these four key elements:
· Establishment of current knowledge of the field;
· Summarize previous research, providing the wider context and background of the importance of the current study;
· Set the stage for the present research, indicating gaps in knowledge and presenting the research question;
· Introduce present research, stating its purpose and outlining its design.
· The authors should establish the current knowledge of the field (forensic) first rather than describing about the mosquitoes first.
d) Materials and methods:
· Please consider adding DNA quantitation as well in header 2.6 since the quantitation method is described in the same header.
· Line 187-189: The authors cited Swaran and Welch [18] when describing about PCR reaction using Power PowerPlex® Fusion System (“…following the instruction manual and as detailed in Swaran and Welch [18]”). Please elaborate on which particular detail because that article was using different multiplex (PowerPlex® 16 HS).
· Please add in the capillary electrophoresis method in the STR profiling section to provide a complete STR analysis.
e) Results:
· The authors mentioned earlier the concentrations were undetectable at 48 h but here the concentrations were expressed up to 5 decimal points. As such would there be any value for 72 h then? Please consider reporting your DNA quantitation according to your limit of detection.
· The authors reported that the concentrations of DNA extracted from control blood samples were significantly higher. This is expected because of very high DNA input was used for controls. Please elaborate why high concentration of DNA control (200 μl of blood sample) was used for extraction, which is not comparable to that of DNA extracted from mosquitos.
· Figure 1 and Table 1 contain repetitive information on DNA concentration. Please consider merging both information into one informative figure/ table. The CT values reported were also not discussed in the manuscript.
· The authors mentioned “..incomplete STR profiles were obtained from mosquito blood meals from the 24 to 36 h post-feeding..” but in the abstract it was written as “Complete and partial STR profiles were obtained up to 24 h and 36 h post-feeding, respectively”. Please clarify.
· Please provide a better resolution of figure 2a. The noise of the baseline is quite obvious in the profile especially in the red dye channel whereby artifacts can be seen clearly. Please provide clarification with regards to this.
· Please provide a better resolution of figure 2c, the STR profile from the blood meal of “G1” (taken from human male blood) at 36 hours post-feeding indicating a partial DNA profile. However, it can be seen that some peaks were not called despite having an interpretable peak. Please provide the analytical threshold for the analysis as it will determine the allele call as the figure is not clear to evaluate.
· Figure 2b and 2d are missing in the manuscript.
· Since the manuscript emphasized on mixed DNA blood meals, figures on mixed DNA profile should be depicted as figures in the manuscript rather than only single DNA profile. The fact that the noise of the baseline is quite obvious in the single male DNA profile, it would be necessary to evaluate the electropherogram of a mixed DNA profile obtained from mosquito blood meals taken from mixed human rather than tabulating the DNA profiles.
· The authors tabulated the DNA profile at the Amelogenin locus (Table 4 and 7) as XXY to indicate mixture between a female and a male. Please refer the nomenclature of reporting Amelogenin locus for mixture as XXY would indicate Klinefelter syndrome (XXY).
· Please provide abbreviation for N/A and please refer the nomenclature guidelines for reporting partial DNA profile in the tables.
· Please indicate whether the STR profiling was carried out for 72 h post-feeding since it was not tabulated and discussed.
f) Discussions:
· The authors mentioned “..STR profiles were obtained up to 36 h post-feeding, after which they became undetectable at 48 h post-feeding.” Results indicated that there are DNA profiles obtained at 48 h post-feeding. Please check for inconsistencies through out the manuscript with regards to DNA results (quantitation as well as STR results) for samples at 48 h.
· Limitations of the study may need to be indicated since it was mentioned earlier by the authors.
· It would be an added value if the authors can address the practice of collecting blood-fed mosquitoes at the scene of crime so that the readers (e.g., crime investigators) would have a better picture of the implementation.
g) Conclusion:
Please consider putting the conclusion in a separate header.

A comprehensive language editing is necessary.
Round 2
Reviewer 1 Report
Again:
1)
Please remove the remaining chatty style. You did it in the abstract but not in the introduction:
"1. Introduction
-
50 Violent crimes are committed all over the world and usually result
-
51 in bloodshed. The accurate detection and confirmation of human blood
-
52 traces and bloodstains obtained from the crime scenes are crucial in forensic
-
53 analysis [1]. The information gained from the correct analysis of bloodstain
-
54 can include what did (or did not) take place, answer the question of who
-
55 may have been involved in these actions, and could be the determining
-
56 factor between guilt and innocence. Blood is usually found in the crime
-
57 scene as dried stains on clothes or solid surfaces such as wood, knife,
-
58 brick....etc which need specific materials and methods for proper collection
-
59 and identification [2]. In comparison, a blood-fed mosquito (if found in the
-
60 crime scene) can provide proper a fresh human blood sample (its blood meal)
-
61 that is suitable for forensic analysis [3]. It is well known that adult female" etc. etc.
- Please see my remarks in the first round of refereeing and change also the introduction.
2)
As already mentioned in the first round, pls remove empty phrases like
→ "It is well known"
→ "It is important to underline"
→ "There is evidence that". Please remove this; please use scientific briefness & clarity without extra words that have no factual meaning = not a chatty style like in a lecture or popular talks.
3)
- Pls. delete "Polymerase chain reaction was performed for amplifying STR loci" because it is clear that this is the only purpose here.
4)
It is hard to judge if the tables and figures are at their right spot in the text but this is most likely due to the editorial system. It also switches from landscape all of a sudden. I will tell the editorial house to take care of it.
5)
english and capital letters are scrambeled:
-
line 407 Previous studies on theAutosomal STR Loci of the U.S. population Data
6)
-
Mixed blood meals taken from 460 humans and animals would contribute to maximizing DNA degradation and 461 then affect STR identification.
→ Why "degradation"? Pls. explain. Degradation refers mostly to fragmentation etc., this is not a matter of one or several blood meals.
7)
Please much shorter, remove empty phrase:
-
"Taken all 382 together, it would be reasonable to strongly support using a mosquito
8)
Please remove "valuable":
-
Insects collected at the crime scene can serve as a valuable source of 359 DNA information in forensic casework [33].
this is an empty phrase, the only question is if you got results (and you did get them, all good)
9)
No, not all are found commonly worldwide, pls be more precise (e.g., some are found worldwide)
-
a) There are over 3600 species of mosquitoes, and they are 373 commonly found worldwide [36];
10)
-
DNA samples in the current study had insufficient
-
399 quantities, which could be attributed due to two possible reasons:
11)
What do you mean by "fewer allele frequencies"? Pls be more precise:
-
human 426 female with mouse blood showed fewer allele frequencies than non-mixed
Thank you for working a little bit more on your paper.
Pls remove the remaining empty phrases.
Author Response
A PDF file attached

Reviewer 2 Report
The authors have addressed most of the problems, but the manuscript still needs some minor changes.
The author frankly admitted that he did not use other kits because of limited budget, which I can understand. However, the authors need to show the limitations in the discussion, so as to enlighten future research.
The authority’s name “Linnaeus” is still not added for “Culex pipiens” in this version. Please add it. Please check the full manuscript carefully and add the authority’s name for other mosquito species.
Author Response
A PDF file attached

Reviewer 3 Report
While the authors have corrected several parts of the manuscript, I found out that they have not addressed the following comments that I made during the earlier review.:
a) Results:
· The authors expressed the concentrations up to 5 decimal points. Please consider reporting your DNA quantitation according to your limit of detection.
· Figure 1 and Table 1 contain repetitive information on DNA concentration. Please consider merging both information into one informative figure/ table with suitable statistical inferences.
· The CT values reported were also not discussed in the manuscript.
· The authors have provided a better resolution of figure 2c; however, it can be seen that some peaks were not called despite having an interpretable peak. Please provide the analytical threshold for the analysis as it will determine the allele call.
b) Discussions:
· Lines 406-407: Change ‘the Autosomal STR Loci of the U.S.population Data’ to ‘the autosomal STR loci of the U.S. population data’.
· Line 408: Change ‘optained’ to ‘obtained’.
· Lines 410-411: The authors mentioned “..STR profiles were obtained up to 36 h post-feeding, after which they became undetectable at 48 h post-feeding.” Results indicated that there are DNA profiles obtained at 48 h post-feeding. Please check for inconsistencies throughout the manuscript with regards to DNA results (quantitation as well as STR results) for samples at 48 h.
It can be improved further. Please correct the grammar and spelling errors.
Author Response
A PDF file attached
